

# The effect of pupil size and peripheral brightness on detection and discrimination performance

Sebastiaan Mathôt[1] and Yavor Ivanov[1,2,3]

[1] Department of Experimental Psychology, University of Groningen, Groningen, Netherlands
[2] Department of Experimental and Applied Psychology, Vrije Universiteit Amsterdam, Amsterdam, Noord Holland, Netherlands
[3] Institute for Brain and Behavior, Amsterdam, Netherlands

## ABSTRACT

It is easier to read dark text on a bright background (positive polarity) than to read bright text on a dark background (negative polarity). This positive-polarity advantage is often linked to pupil size: A bright background induces small pupils, which in turn increases visual acuity. Here we report that pupil size, when manipulated through peripheral brightness, has qualitatively different effects on discrimination of fine stimuli in central vision and detection of faint stimuli in peripheral vision. Small pupils are associated with improved discrimination performance, consistent with the positive-polarity advantage, but only for very small stimuli that are at the threshold of visual acuity. In contrast, large pupils are associated with improved detection performance. These results are likely due to two pupil-size related factors: Small pupils increase visual acuity, which improves discrimination of fine stimuli; and large pupils increase light influx, which improves detection of faint stimuli. Light scatter is likely also a contributing factor: When a display is bright, light scatter creates a diffuse veil of retinal illumination that reduces perceived image contrast, thus impairing detection performance. We further found that pupil size was larger during the detection task than during the discrimination task, even though both tasks were equally difficult and similar in visual input; this suggests that the pupil may automatically assume an optimal size for the current task. Our results may explain why pupils dilate in response to arousal: This may reflect an increased emphasis on detection of unpredictable danger, which is crucially important in many situations that are characterized by high levels of arousal. Finally, we discuss the implications of our results for the ergonomics of display design.

## INTRODUCTION

You are probably reading this text as dark letters on a bright background. And if not, then you might consider doing so, because it is easier to read dark letters on a bright background (positive polarity) than it is to read bright letters on a dark background (negative polarity). This positive-polarity advantage has been well-established in human-factors research (*Buchner, Mayr & Brandt, 2009*; *Dobres, Chahine & Reimer, 2017*; *Piepenbrock, Mayr &*

Corresponding author
Sebastiaan Mathôt, s.mathot@rug.nl

*Buchner, 2014b*; *Piepenbrock, Mayr & Buchner, 2014a*; *Taptagaporn & Saito, 1990*), and is often studied using proofreading experiments. For example, *Piepenbrock, Mayr & Buchner (2014b)* asked participants to verbally report all misspelled words in a short text. The authors found that participants read faster, and spotted more mistakes, when the text was presented in a positive polarity, as compared to a negative polarity. Findings such as these are among the reasons that most websites and word-processing software use a positive polarity.

Several researchers have suggested that the positive-polarity advantage is linked to pupil size (e.g., *Piepenbrock, Mayr & Buchner, 2014b*). When the background of a display is bright, the pupil constricts, compared to when the background is dark; this is the pupil light response (reviewed in (*Mathôt, 2018*; *Mathôt & Van der Stigchel, 2015*). In terms of visual perception, there are three main consequences of a bright background and the resulting pupil constriction. The first consequence is negative: A bright background, as any source of brightness, results in light scatter; that is, some of the incoming light is not focused, but instead spreads over a large part of the retina. This results in a diffuse veil of light that reduces perceived image contrast. The second consequence is also negative: Small pupils reduce the amount of light that falls on the retina, and thus reduce the signal-to-noise ratio of the image. The third consequence is positive: Small pupils suffer less from optical distortions that reduce image quality, and thus increase visual acuity (*Campbell & Gregory, 1960*; *Liang & Williams, 1997*; *Lombardo & Lombardo, 2010*; *Woodhouse, 1975*); that is, small pupils result in sharper vision. (Background luminance also affects other aspects of vision, such as contrast sensitivity, in complex ways, as reviewed by *Kalloniatis & Luu (1995)*. Whether, and if so how, these factors also play a role in the positive-polarity advantage is not yet fully clear.)

When reading text that is presented with sufficiently high contrast, as is typically the case in daily life, the advantage of increased visual acuity seems to outweigh the disadvantages of reduced signal-to-noise ratio and increased light scatter. Therefore, it is easier to read dark text on a bright background (when pupils are small), especially when the text is written in a small font (*Piepenbrock, Mayr & Buchner, 2014a*).

There is also some neurophysiological evidence that small pupils increase visual acuity. For example, Bombeke and colleagues (*2016*) manipulated pupil size by having participants covertly attend to either a bright or a dark disk, which respectively constricts or dilates the pupils, without changing eye position or visual input (cf. *Binda, Pereverzeva & Murray, 2013*; *Mathôt et al., 2013*). They then briefly presented a task-irrelevant but salient line-grating stimulus in peripheral vision, and measured the C1, an event-related potential (ERP) component that is associated with activity in primary visual cortex. The amplitude of the C1 was larger when participants covertly attended the bright disk (resulting in small pupils), compared to when they covertly attended the dark disk (resulting in large pupils). According to the authors, this result was due to the fact that small pupils improved the resolution of the C1-eliciting stimulus, in turn leading to a stronger neural response in primary visual cortex.

However, behavioral evidence for a link between pupil size and visual acuity is mixed. In a recent study by *Ajasse, Benosman & Lorenceau (2018)*, participants made a sequence

of eye movements toward a configuration of disks; each disk had a different brightness, and the size of the pupil was therefore different depending on which disk the participant was fixating. While participants were fixating a disk, two gabor patches were briefly and sequentially presented in their visual periphery. The spatial frequency of the gabor patches differed, and participants indicated which of the two had the highest spatial frequency. The authors predicted that performance on this task should increase with decreasing pupil size (and thus with increasing brightness of the fixated disk). However, they found no such relationship; that is, performance did *not* depend on pupil size.

The results of Ajasse and colleagues (*2018*) show that small pupils do not lead to improved discrimination performance in every situation. Specifically, in their experiment, stimuli were presented in peripheral vision, where acuity is mostly limited by the reduced density of cone photoreceptors; therefore, in peripheral vision, optical blur due to large pupils likely has at most a very small effect on stimulus discrimination. However, the results of Bombeke and colleagues (*2016*), who also used a peripherally presented stimulus, suggest that under specific conditions a small-pupil advantage can also be found in peripheral vision.

In yet other situations, small pupils may even *impair* visual performance (reviewed in *Mathôt, 2018*; *Mathôt & Van der Stigchel, 2015*). Specifically, detecting faint stimuli in peripheral vision requires a high signal-to-noise ratio of the stimulus relative to the background, and visual acuity is only of secondary importance. In this case, large pupils may improve the signal-to-noise ratio of vision by increasing overall light influx; more specifically, when your aim is to detect a target stimulus against a background, then large pupils will enhance the signal strength of both the target and the background, thereby reducing the influence of noise, in turn making the target more easily distinguishable from the background. Therefore, stimulus detection in the visual periphery should benefit from large pupils. When large pupils are associated with a dark environment, as is typically the case in real life, this benefit should be even stronger, because the increased signal-to-noise ratio due to large pupils is accompanied by reduced light scatter due to the dark environment. (A subtle, additional complexity comes from the fact that pupil size and light scatter are likely interdependent such that large pupils increase light scatter by increasing light influx. This may be another reason why large pupils are most advantageous in a dark environment, in which light scatter plays a relatively minor role.)

However, a study by *Thigpen, Bradley & Keil (2018)* suggests that large pupils may not necessarily 'boost' neural responses to visual input. In their study, they presented a rapidly flickering stimulus, and measured so-called Steady-State Visually Evoked Potentials (ssVEPs): neural oscillations in visual cortex with the same frequency as the inducing stimulus. ssVEP power is believed to reflect the level of neural activity. Crucially, the authors found no relationship between ssVEP power and pupil size, and they interpreted this result as evidence for divisive normalization (*Carandini & Heeger, 2012*); that is, they suggested that visual responses, even in early visual cortex, are invariant to overall light influx and thus unaffected by pupil size.

Taken together, previous research has provided compelling evidence for an advantage of small pupils (and a bright background) for text reading (*Buchner, Mayr & Brandt, 2009*;

*Dobres, Chahine & Reimer, 2017*; *Piepenbrock, Mayr & Buchner, 2014b*; *Piepenbrock, Mayr & Buchner, 2014a*; *Taptagaporn & Saito, 1990*). There is also some neurophysiological evidence for a small-pupil benefit for visual acuity (*Bombeke et al., 2016*; but see *Ajasse, Benosman & Lorenceau, 2018*). In contrast, there is no evidence for a large-pupil advantage for stimulus detection (e.g., *Thigpen, Bradley & Keil, 2018*). Nevertheless, a large-pupil advantage for detection is clearly predicted based on the optical properties of the eye (see *Mathôt, 2018*; *Mathôt & Van der Stigchel, 2015*).

The aim of the current study is to demonstrate both a small-pupil advantage for discrimination of stimuli in central vision, and a large-pupil advantage for detection of stimuli in peripheral vision. We will manipulate pupil size by manipulating the brightness of the visual periphery (which likely also affects performance independently of pupil size, as discussed above), while presenting all task-relevant stimuli on a central gray disk of constant brightness.

# EXPERIMENTS 1 AND 2

The goal of Experiments 1 and 2 was to investigate whether pupil size, when manipulated through peripheral brightness, differentially affects performance on detection and discrimination tasks. In Experiment 1, participants detected, or discriminated the orientation of, a tilted Gabor patch. In Experiment 2, participants detected, or discriminated the lexicality of, a single word.

## Methods

### Experiment 1

*Participants, ethics, and apparatus* Nine naive observers participated in the experiment, after providing written informed consent. The experiment was approved by the local ethics committee of Groningen University (16163-SP-NE and 16349-S-NE). Observers were recruited from the community of Groningen University, but for privacy reasons we did not collect specific demographic information. Pupil size was recorded with an EyeLink 1,000 (SR Research). Stimuli were presented with OpenSesame 3.1 (*Mathôt, Schreij & Theeuwes, 2012*) on a 27" flat screen Iiyama monitor with a resolution of 1920× 1,080 px. The monitor was not gamma calibrated. Participants were seated with their head on a chin rest about 50 cm away from the monitor.

*Pupil-size manipulation* Pupil size was manipulated by varying the brightness of the visual periphery (low: 0.16 cd/m$^2$, medium: 8.30 cd/m$^2$, high: 52.26 cd/m$^2$; see Fig. 1), which corresponded to the full display (49.22° × 27.70°) except for a central gray disk. All task-relevant stimuli were presented on the central gray disk (2.84 cd/m$^2$; diameter: 25.65°) that was kept constant throughout the experiment.

*Design* The experimental task (discrimination or detection) was varied between sessions. One experimental session consisted of five blocks.

The first two blocks of each session served to calibrate a Quest adaptive procedure, which varied the properties of the target stimulus (see below) such that accuracy was

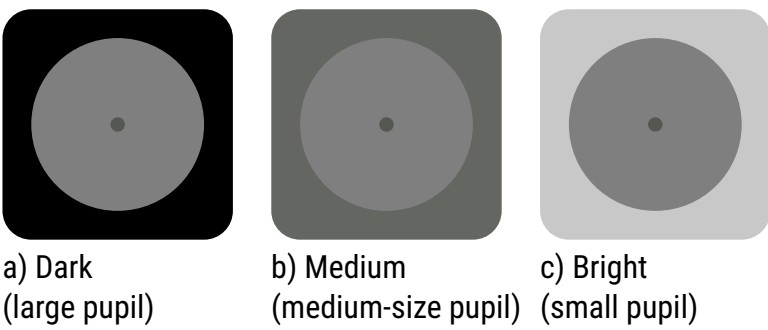

a) Dark
(large pupil)

b) Medium
(medium-size pupil)

c) Bright
(small pupil)

**Figure 1** **Luminance manipulation in Experiments 1 and 2.** (A) Dark (large pupil); (B) Medium (medium-size pupil); (C) Bright (small pupil). The luminance of the visual periphery was varied to manipulate pupil size. All task-relevant stimuli appeared on a central gray disk that was kept constant throughout the experiment.

kept at 75%. During these calibration blocks, peripheral brightness was set to 2.84 cd/m$^2$. After these two blocks, the Quest procedure was stopped, and the target was kept constant throughout the remainder of the session. Next, participants performed three blocks of 50 trials. Peripheral brightness was varied between blocks (Fig. 1).

Block order was fully counterbalanced, such that each possible order occurred once for each participant and task. Half the participants started on the first day with a discrimination session followed by a detection session, vice versa on the second day, etc. The other half of the participants started with a detection session on the first day. In some cases, participants did more than two sessions per day. In total, participants performed 3,000 trials across 12 sessions in approximately six hours.

*Discrimination task* In the discrimination task (see Fig. 2A), each trial started with a central fixation dot (a uniform patch with a gaussian envelope with a standard deviation of 0.51° [20 px] and a peak brightness of 4.41 cd/m$^2$) that was removed after 500 ms. After a random interval between 500 and 1,500 ms, drawn from a uniform distribution, a central target stimulus linearly faded in and out over a period of 650 ms. The target was a centrally presented sinusoidal grating with a gaussian envelope (a Gabor patch) with a standard deviation of 0.51° (20 px). To maintain accuracy at 75%, the spatial frequency (range = [7.8, 11.7] cycles/°), contrast (range = [1, 25] %), and orientation (range = [1, 2.5]°) of the target was varied with a Quest adaptive procedure during calibration blocks as described above.

At any point during the trial, participants pressed the left arrow key if the target was tilted counterclockwise from a vertical orientation, and the right arrow key if it was tilted clockwise (i.e., a two-alternative forced choice). The trial ended 3 s after the onset of the target. If the participant did not press any key within 3 s, a timeout was registered, and the response was considered incorrect.

*Detection task* In the detection task (see Fig. 2B), each trial started with a central fixation dot (4.41 cd/m$^2$) that remained visible throughout the trial. On 50% of trials, after a

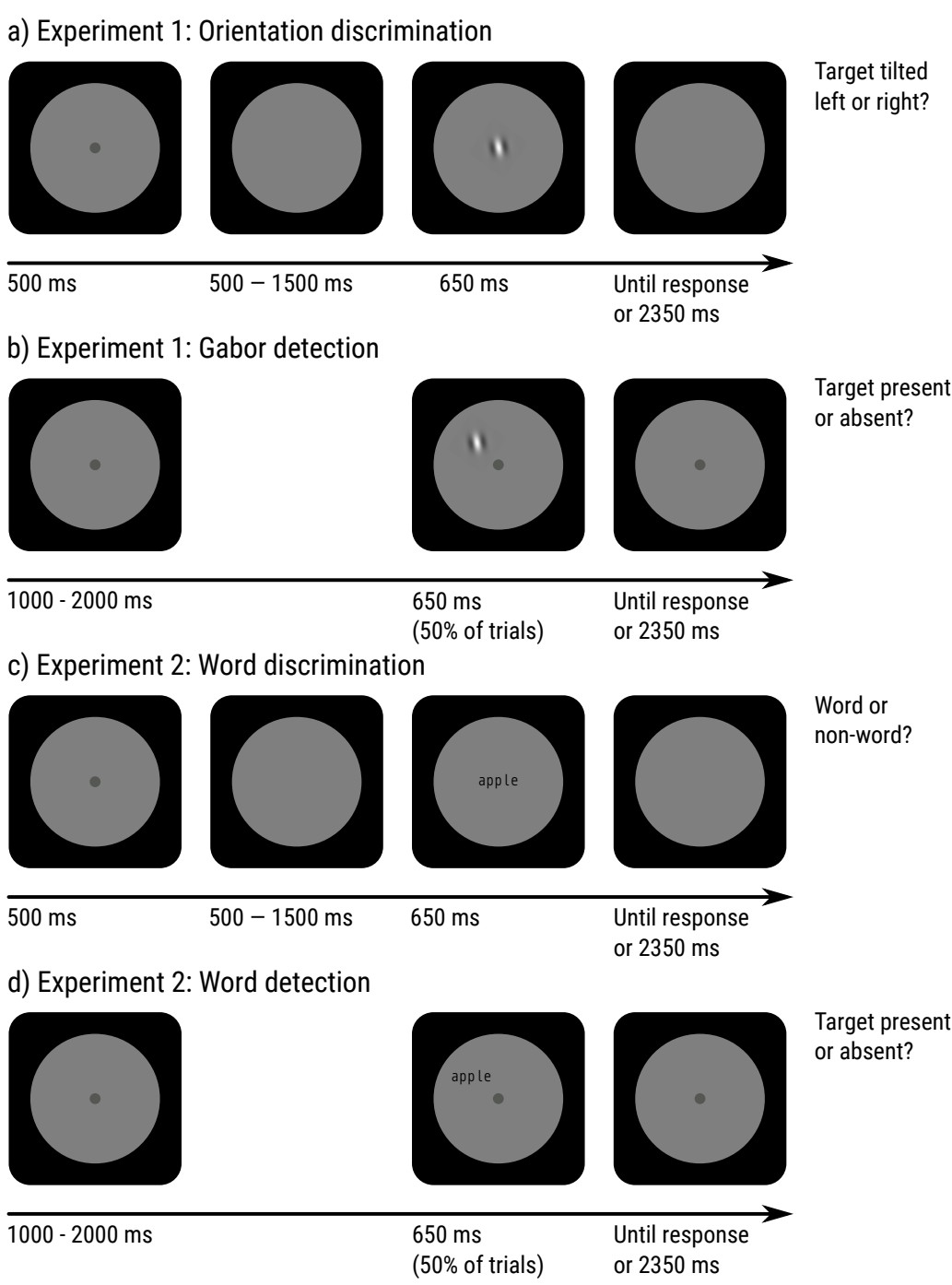

**Figure 2 Schematic paradigm of Experiments 1 and 2.** (A) Orientation-discrimination task for Experiment 1. (B) Orientation-detection task for Experiment 1. (C) Word-discrimination (lexical decision) task for Experiment 2. (D) Word-detection task for Experiment 2.

random period drawn from a flat distribution between 1 and 2 s, a target stimulus was linearly faded in and out over a period of 650 ms. The target was identical to that of the discrimination task, except that its standard deviation was 1.02° (40 px; i.e., twice as big), that the spatial frequency was twice as low (range = [3.9, 5.9] cycles/ °), and that it was presented at a random point on an imaginary circle around the fixation dot with a radius of 7.70° (300 px).

At any point during the trial, participants pressed the space bar when they detected a target, and did not press any key when they did not detect a target. The trial ended 3 s after the onset of a target (when present), or after a random interval between 4 and 5 s, drawn from a uniform distribution.

### Experiment 2

Experiment 2 was in most ways identical to Experiment 1, and only the differences are described below.

*Participants, ethics, and apparatus*  Nine naive observers, most of whom had not participated in Experiment 1, participated in the experiment after providing written informed consent. All participants were native Dutch speakers.

*Stimulus selection*  We selected the 750 most highly frequent words between four and six characters from the Dutch Lexicon Project (*Keuleers, Diependaele & Brysbaert, 2010*), after manually (and based on our subjective impression) excluding overly offensive words. For each word, a matching pseudoword was generated with Wuggy (*Keuleers & Brysbaert, 2010*).

*Design*  Participants performed 1,500 trials across six sessions in approximately three hours. All participants saw all words and pseudowords once in a random order.

*Task*  Targets were (pseudo)words presented in a monospace font (Droid Sans Mono). In the discrimination task, targets were centrally presented, and participants pressed the left arrow key if the target was a pseudoword and the right arrow key if it was a word (i.e., a lexical-decision task). In the detection task, targets were peripherally presented on 50% of trials, and participants pressed the space bar if they detected a target, and did not press any key otherwise. To maintain accuracy at 75%, the font size and contrast of the target was varied.

## Results

We performed the same set of analyses on both experiments. The results from both experiments were very similar.

### Task performance

To be able to directly compare performance in the detection and discrimination tasks, we used accuracy (the percentage of correct responses) as our dependent measure. However, the results for the detection task are similar when using $d'$ (a measure of sensitivity that is based on signal-detection theory). Mean accuracy on the detection task was 74.7% (Exp

1) and 73.2% (Exp 2). Mean accuracy on the discrimination task was 74.5% (Exp 1) and 75.2% (Exp 2).

To test whether pupil size (as manipulated through peripheral brightness) affects performance (see Fig. 3), and does so differently for the discrimination and detection tasks, we conducted a generalized linear mixed effects model (GLM) with response-correct as dependent variable (binomial), brightness (low [reference], medium, high), condition (detection [reference], discrimination), and the brightness × condition interaction as fixed effects. We included only by-participant random intercepts, because more complex models failed to converge. (However, the results do not crucially depend on the exact model structure.) All mixed-effects analyses were conducted with the R package lme4 (*Douglas et al., 2015*), in which the above model corresponds to the following formula: correct ~brightness * condition + (1|subject_nr).

There was an effect of brightness (Exp 1: $Z = -8.754$, $p < .001$; Exp 2: $Z = -8.005$, $p < .001$), indicating that for the detection (reference) condition, accuracy decreased with increasing brightness. There was an effect of condition (Exp 1: $Z = -5.343$, $p < .001$; Exp 2: $Z = -2.001$, $p = .045$) indicating that for the low (reference) brightness, accuracy was lower for the discrimination than detection condition. Crucially, there was also a brightness × condition interaction (Exp 1: $Z = 6.586$, $p < .001$; Exp 2: $Z = 4.114$, $p < .001$), indicating that the effect of brightness was driven by the detection condition, and not present in the discrimination condition.

To confirm this, we also analyzed the discrimination condition separately, in a model with only brightness as fixed effect. Here we found no effect of brightness in Exp 1 ($Z = 0.503$, $p = .615$), and only a weak effect of brightness in Exp 2 ($Z = -2.153$, $p = 0.031$).

### Pupil size

The EyeLink provides pupil size in arbitrary units. To convert these units to millimeters of diameter, we first recorded artificial pupils (black circles printed on white paper) of different sizes, and then determined a function to convert arbitrary EyeLink pupil units (au) to pupil diameter (mm): $mm = -0.0324 + 0.1075 \times au^{0.5}$

Mean pupil size during the detection task was 4.4 mm (Exp 1) and 5.1 mm (Exp 2). Mean pupil size on the discrimination task was 4.3 mm (Exp 1) and 5.0 mm (Exp 2).

Our brightness manipulation should have a large effect on pupil size. It is also possible that the task affects pupil size, despite the fact that the two tasks were equally difficult. To test this, we conducted a linear mixed-effects analysis (LMER) with pupil size (prior to the onset of the trial) as dependent measure and brightness, condition, and a brightness × condition interaction as fixed effects. Again, we included only by-participant random intercepts, because more complex models failed to converge (see Fig. 4).

There was an effect of brightness (Exp 1: $t = -87.405$; $p < .001$ Exp 2: $t = -58.165$, $p < .001$), reflecting that pupil size decreased with increasing brightness. There was also an effect of condition (Exp 1: $t = -3.849$, $p < .001$; Exp. 2: $t = -4.340$, $p < .001$), reflecting that pupil size was larger in the detection than in the discrimination condition. There

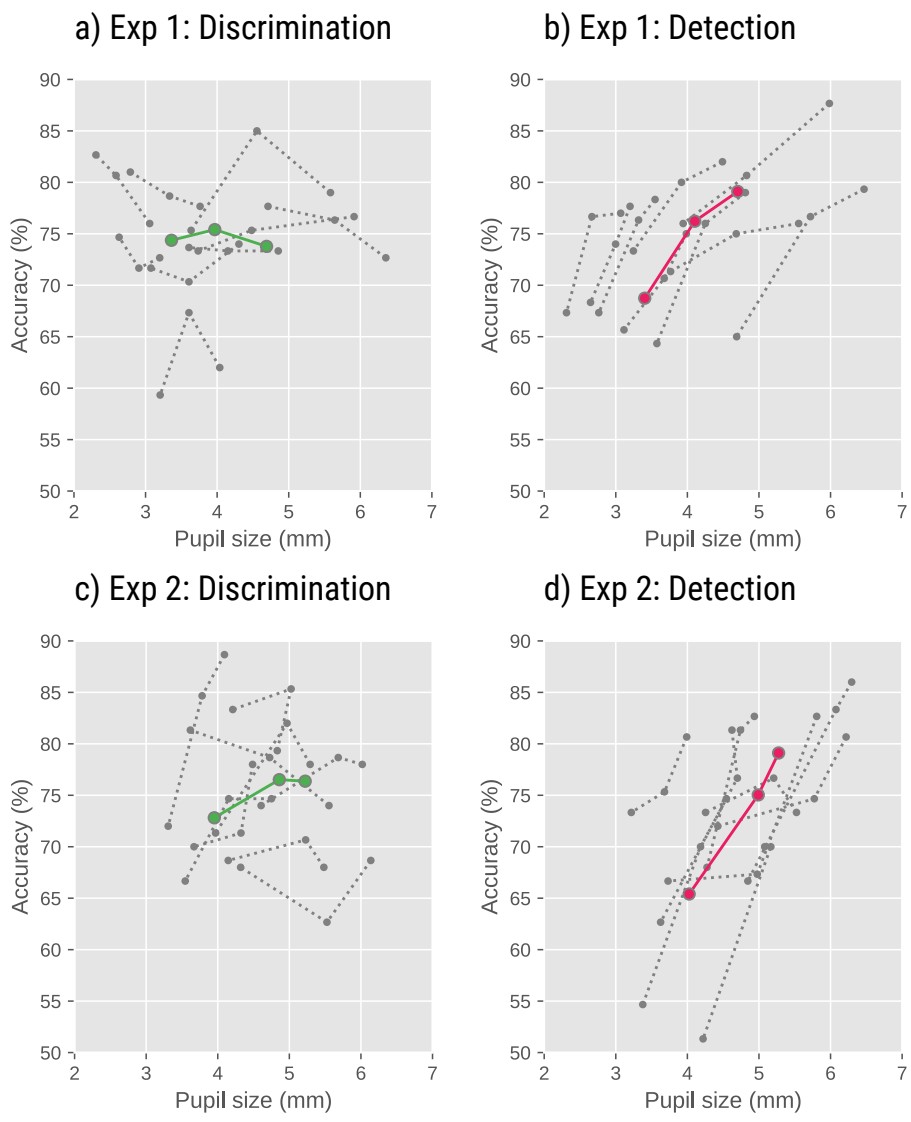

**Figure 3** **Task performance as a function of pupil size in Experiments 1 and 2.** Detection accuracy increased with decreasing peripheral brightness, and thus increasing pupil size (B, D; pink lines). However, there was no effect of peripheral brightness on discrimination performance (A, C; green lines). Gray dotted lines indicate individual participants. Colored solid lines indicate grand averages.

was no notable brightness × condition interaction (Exp 1: $t = -0.930$, $p = 0.352$; Exp 2: $t = -0.216$, $p = 0.829$).

## Discussion

In summary, we found that detection performance was better with large pupils (and a dark periphery) than with small pupils (and a bright periphery). This relationship was strong, robust, and in the direction that we predicted. However, and unlike we predicted, we did not find that discrimination performance increased with decreasing pupil size (and

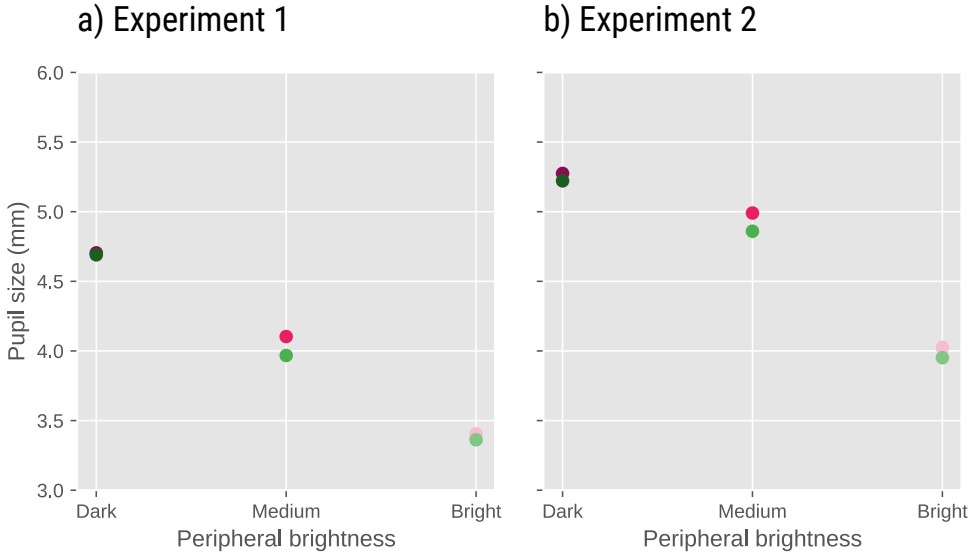

**Figure 4  Pupil size as a function of task and peripheral brightness in Experiments 1 (A) and 2 (B).** In both experiments, pupil size decreased with decreasing peripheral brightness. In addition, pupil size was slightly larger in the Detection (pink dots) than in the Discrimination (green dots) condition.

thus increasing peripheral brightness); in fact, there was a slight tendency in the opposite direction for Exp. 2.

In addition, we found that pupil size was larger in the detection than in the discrimination condition, even though both tasks were equally difficult.

A limitation of our setup for measuring discrimination performance was that we could not present very small stimuli: When presented at full contrast, even the finest possible grating (i.e., 2 px/cycle) or the smallest possible letter (5× 5 pixels) could be discriminated without too much trouble by someone with normal vision. Therefore, to increase the difficulty of the discrimination task, we also reduced the contrast of the target stimulus, and our discrimination task was therefore not a pure measure of discrimination performance (a limitation that we addressed in Experiment 3).

## EXPERIMENT 3

In Experiment 3, we used a setup that allowed us to present very small letters. The aim of this experiment was to investigate whether we could observe an advantage of small pupils (and increased peripheral brightness) on discrimination performance in a task that tested the limits of visual acuity. If so, this would suggest that the absence of a small-pupil benefit in Experiments 1 and 2 was due to the fact that, in these experiments, our stimuli were not sufficiently fine to test the limits of visual acuity.
## a) Experimental set-up

E — Upper- or lowercase?

## b) Brightness manipulation

**Figure 5** Schematic set-up and paradigm of Experiment 3. (A) Participants indicated whether a target letter was uppercase or lowercase. (B) Pupil size was manipulated by varying the brightness of two displays that were positioned near the participant, and flanked the target display.

## Methods

### Participants, ethics, and apparatus

20 naive observers participated in the experiment, after providing informed consent. The experiment was approved by the local ethics committee of Groningen University (16163-SP-NE and 16349-S-NE). Observers were recruited from the community of Groningen University, but for privacy reasons we did not collect specific demographic information. Pupil size was recorded with an EyeLink 1,000 (SR Research). Stimuli were presented with OpenSesame 3.1 (*Mathôt, Schreij & Theeuwes, 2012*) on three separate 7" tablets (Samsung Galaxy Tab 7), each with a resolution of $1280 \times 800$ px. Two tablets were presented nearby, at angle of about 50° (relative to the direction of gaze) and a distance of about 50 cm on both sides of the participant's head, and served as light sources (see Fig. 5A). One tablet was placed in front of the participant, at a distance of 5.5 m, and served as the target display.

### Task and design

Each trial started with the presentation of a black central fixation dot ($R = 0.11°$ [90 px]) for 500 ms on the target display. Next, a lowercase or uppercase letter ('a', 'b', 'd', 'e', 'g', 'h', 'l', 'm', 'n', 'q', 'r', or 't') was presented for 2.5 s. Letters were presented centrally in black monospace font (Droid Sans Mono). The size of the letters was varied with a Quest adaptive procedure to converge on 75% accuracy; this resulted in a range of vertical letter sizes between roughly 0.25° and 0.08° (visual degrees). Participants pressed the 'z' key to indicate that the letter was lowercase, and the '/' key to indicate that the letter was uppercase.

The experiment started with a practice block of 75 trials during which the background of all tablets was gray (72.15 cd/m$^2$ for target display; 69.01, 64.70 cd/m$^2$ for peripheral displays). This practice block also served to determine an appropriate font size to start with during the experimental blocks. Next, participants performed six experimental blocks during which the brightness of the light-source tablets was varied (343.30 cd/m$^2$ and 345.20 cd/m$^2$ for the two peripheral displays], medium [69.01 cd/m$^2$, 64.70 cd/m$^2$], or dark [0.83 cd/m$^2$, 0.88 cd/m$^2$]) while the background of the target display remained gray (see Fig. 5B). Each experimental block started with the font size that the practice block had ended with. The order of the experimental blocks followed a counterbalanced ABCABC design. In total, participants performed 375 trials in approximately 40 min.

## Results

### Task performance

For each participant and brightness level separately, we took the final Quest test value as a measure of performance (font size was determined based on the Quest value, and the final Quest value corresponded to a font size resulting in 75% accuracy). To test whether peripheral brightness (and pupil size) affects performance, we conducted a linear mixed effects model (LME) with final Quest value as dependent measure and brightness as fixed effect. We included by-participant random intercepts and slopes. There was an effect of brightness ($t = -2.356$, $p = .029$), indicating that discrimination performance increased with increasing peripheral brightness (Fig. 6).

If pupil size affects performance, then we should not only find differences between the conditions in which we experimentally manipulate pupil size, but also between participants that differ in the size of their pupils. The fact that we had increased the number of participants from 9 (in Exp 1 and 2) to 20 (in Exp 3) made it feasible to look at whether individual differences in pupil size affect performance. To do so, we determined, for each participant separately, the mean pupil size during the entire experiment, and the average final Quest value (averaged across brightness levels). There was a correlation between the two measures ($r = -.508$, $p = .022$), indicating that participants with smaller overall pupils had higher performance; however, this correlation was largely driven by one participant with especially large pupils, and after excluding this participant the correlation was no longer reliable ($r = -.259$, $p = .284$).

### Pupil size

To test whether our brightness manipulation affects pupil size, we conducted an LME with pupil size as dependent measure and brightness as fixed effect. We included by-participant random intercepts and slopes. There was an effect of brightness ($t = -12.662$, $p < .001$), indicating that pupil size decreased with increasing brightness of the flanking tablets. Pupil size was converted from arbitrary units to millimeters of diameter with the same procedure as used for Experiments 1 and 2.

## Discussion

As predicted, we found that small pupils (and increased peripheral brightness) improved discrimination performance. In, addition we found some evidence that participants with

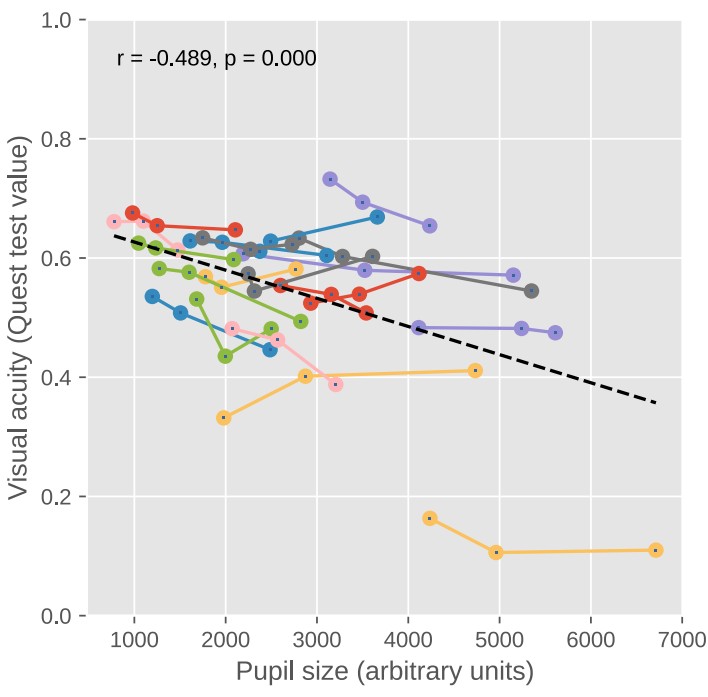

**Figure 6** **Visual acuity as a function of pupil size in Experiment 3.** Discrimination performance increased with increasing peripheral brightness (and decreasing pupil size). Lines correspond to individual participants (for clarity, different participants have different colors). Dots correspond to different levels of peripheral brightness, such that the highest peripheral brightness corresponds to the smallest pupil size. In addition, participants with smaller overall pupils had higher discrimination performance (although this effect was largely driven by one participant with very large pupils). The dashed line indicates the regression line.

small pupils had higher discrimination performance, although this result was driven largely by a single participant with very large pupils. That is, there was a clear link between pupil size and discrimination performance, at least when pupil size was manipulated experimentally, and possibly also when considering individual differences.

## GENERAL DISCUSSION

Here we report that pupil size, when manipulated through peripheral brightness, has qualitatively different effects on discrimination of fine stimuli in central vision and detection of faint stimuli in peripheral vision.

Specifically, we found that small pupils (and thus a bright periphery) are associated with improved discrimination of small letters presented in central vision. This is consistent with previous studies that showed a so-called positive-polarity advantage; that is, it is easier to read dark letters on a bright background (positive polarity) than it is to read bright letters on dark background (negative polarity) (*Buchner, Mayr & Brandt, 2009*; *Dobres, Chahine & Reimer, 2017*; *Piepenbrock, Mayr & Buchner, 2014b*; *Piepenbrock, Mayr & Buchner, 2014a*; *Taptagaporn & Saito, 1990*). We observed this association only (but highly reliably) with very small letters that were at the limits of visual acuity. This is consistent with a previous

study showing that the positive-polarity advantage is most pronounced for small letters (*Piepenbrock, Mayr & Buchner, 2014a*). The small-pupil benefit for discrimination is likely due to the fact that visual acuity is highest with small pupils, which suffer less from optical distortions that reduce visual acuity (*Campbell & Gregory, 1960*; *Liang & Williams, 1997*; *Lombardo & Lombardo, 2010*; *Woodhouse, 1975*; *Bombeke et al., 2016*).

We further found that large pupils (and thus a dark periphery) are associated with improved detection of faint stimuli that were presented at an unpredictable location in peripheral vision. This large-pupil benefit for detection is likely due to two factors. First, large pupils increase light influx, which increases the signal-to-noise ratio of vision, which in turn facilitates detection of very faint stimuli (but perhaps not, or hardly, of stimuli that are presented well above the detection threshold, as used for example by *Thigpen, Bradley & Keil (2018)*). Second, the dark periphery that we used to induce large pupils resulted in reduced light scatter (*Lombardo & Lombardo, 2010*), in turn resulting in increased image contrast, thus making it easier to detect stimuli. Therefore, reduced light scatter likely also contributed to the large-pupil benefit (which is therefore in part likely a dark-periphery benefit).

Our results offer a possible explanation for why pupils dilate in response to increased arousal (e.g., *Mathôt, 2018*; *Mathôt & Van der Stigchel, 2015*). Situations that require fine discrimination are often characterized by low levels of arousal, and situations that require detection are often characterized by high levels of arousal. For example, arousal is low when a person is reading a book, or when an animal is foraging for food. In such cases, it is crucially important to identify what you're looking at. In contrast, arousal is high when a person is afraid, or when an animal is on the lookout for predators. In such cases, it is crucially important to detect unexpected dangers. In other words, pupil dilation in response to arousal may reflect an increased emphasis on visual sensitivity, at the expense of visual acuity, to meet the demands of the situation.

An incidental yet striking result is that pupil size was larger during the detection task than during the discrimination task. Because there was no systematic difference in difficulty between the two tasks, this pupil-size difference is likely not due to differences in mental effort (which is known to affect pupil size, see e.g., *Mathôt, 2018*). One possibility is that, in the detection task, the pupil dilated as a result of attention being directed to peripheral rather than central vision (cf. *Brocher et al., 2018*; *Daniels et al., 2012*). An even more interesting possibility is that the pupil automatically assumes a size that is optimal for the current task, and that arousal-related pupil responses are merely one example of this general principle.

Our results also have implications for the ergonomics of display design. The idea that information is best presented as dark stimuli against a bright background (positive polarity) is well-established within human-factors research (*Buchner, Mayr & Brandt, 2009*; *Dobres, Chahine & Reimer, 2017*; *Piepenbrock, Mayr & Buchner, 2014b*; *Piepenbrock, Mayr & Buchner, 2014a*; *Taptagaporn & Saito, 1990*). Our results confirm this idea, but add the important caveat that this may only hold for displays that contain highly detailed information that should be discriminated; an example of such a display would be a web page that contains text in a small font. In contrast, displays that emphasize detection

over discrimination may function best when bright stimuli are presented against a dark background (negative polarity); an example of such a display might be an air-traffic control display. More generally, whether a positive or a negative polarity is best depends on many factors, some of which we have highlighted in the present study. But there are many additional factors that we have not considered here, including the role of light adaptation, discomfort glare, contrast sensitivity, and the extent to which central or peripheral vision is used (*Kalloniatis & Luu, 1995*).

Finally, our results also speak to the recent debate about whether pupil size has been a confound in previous studies in the field of cognitive neuroscience. Bombeke and colleagues (*2016*) have argued that pupil size affects early visual processing; this implies that whenever an experimental manipulation affects pupil size, any difference in brain responses may be due to differences in pupil size. In contrast, Thigpen and colleagues (*2018*) have argued that differences in pupil size do not markedly affect early visual processing; this implies that researchers can safely interpret differences in brain responses between conditions, even if these are accompanied by differences in pupil size. Our own position is somewhere in between. It seems that differences in pupil size can measurably affect behavior and, since behavior originates from the brain, also brain activity. However, we induced pupil-size differences that were far larger than the few-percent change that is generally observed in cognitive-neuroscience experiments. And even so, we had to go through great lengths in order to find a measurable effect on visual acuity. In other words, in most previous experiments the confounding effect of pupil size has likely been small.

In summary, we have shown that small pupils, induced through a bright periphery, are associated with improved discrimination of fine stimuli in central vision. In contrast, large pupils, induced through a dark periphery, are associated with improved detection of faint stimuli in peripheral vision.

### Funding
This work was supported by the Dutch organization for scientific research (NWO), Grant 016.Veni.175.078 award to SM The funders had no role in study design, data collection and analysis, decision to publish, or preparation of the manuscript.

### Grant Disclosures
The following grant information was disclosed by the authors:
Dutch organization for scientific research (NWO): 016.Veni.175.078.

### Competing Interests
The authors declare there are no competing interests.

### Author Contributions
- Sebastiaan Mathôt conceived and designed the experiments, analyzed the data, prepared figures and/or tables, authored or reviewed drafts of the paper, approved the final draft.

- Yavor Ivanov conceived and designed the experiments, performed the experiments, authored or reviewed drafts of the paper, approved the final draft.

## Human Ethics

The following information was supplied relating to ethical approvals (i.e., approving body and any reference numbers):

The experiment was approved by the local ethics committee of Groningen University (16163-SP-NE and 16349-S-NE).

## Data Availability

The raw data is available at: Mathôt, S., and Ivanov, Y. 2019. "Detection, Discrimination, and Pupil Size." OSF. July 8. http://osf.io/h389s.

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
