# Peer review of "The effect of pupil size and peripheral brightness on detection and discrimination performance"

_PeerJ, doi:10.7717/peerj.8220_

## Round 0.1 · original submission · Major Revisions

Each reviewer identified significant strengths associated with the work, however, there were a number of issues which require attention- In particular, Figures 4 and 5 need to be referenced in the main body text, and their importance for the overall arguments clearly described.
Each reviewer also pointed out a number of analytical and statistical clarifications and/or recommendations for improvement which should be attended to. Reviewer 2 suggests a pharmacological experiment (phenylephrine pupil dilation), which is intriguing, but which I believe is outside the scope of this work, and which would require additional planning and Ethics approval. Thus, while you may consider exploring this idea as a "Future Direction for research", this additional study will not be required for consideration of this work for publication.
In your revision, and in the resubmission letter, please indicate the specific changes or analyses you have implemented to address the remaining concerns identified by each reviewer. Where multiple reviewers have identified the same concern, it is sufficient to address the point only once in the letter.
Please note that the revised manuscript will be subject to re-review prior to a final publication decision.
I look forward to reading the revised document, and I thank you again for choosing to submit your work to PeerJ.

·

Basic reporting

The introduction explores the relevant literature clearly and in an appropriate level of detail. The paper itself is self contained with relevant results to address the hypotheses.

Overall the English is very good, with a couple of small exceptions:
1) most importantly, be very clear about the stages of light transmission, image formation on the retina, and processing/perception of the image ("vision"). There are some strange constructions that are produced by muddiness here.

line 50: "... veil of light that reduces [effective] image contrast", I suggest adding "effective" because of course the physical stimulus is unchanged.

line 54-55: "small pupils see sharper", I suggest omitting this clause entirely. Pupils don't "see" at all, and you've explained how size affects image formation already.

line 66: "participants attended the bright disk", I suggest replacing "attended" with "fixated", if this is appropriate. Attended allows for the possibility of covert attention, which may be shifted to different parts of the visual image without changing the input to the retina.

line 85/86: "a high signal-to-noise ratio of the image", this is confusing, because I think you mean high SNR of image relative to background, not the SNR intrinsic to the image (although these quantities are correlated in many cases). If my interpretation is correct, I suggest adding "relative to background" or something similar to clarify.

2) Several nouns that are semantically important to the structure of the study are capitalised throughout ("Detection", "Target", "Discrimination", etc...) this is not standard English, and capitals should be removed.

3) Please interpret the outputs of the Quest algorithm. Have you configured it to find a 75% threshold? i.e. "Final Quest Value" should be interpreted as the font size threshold: an estimate of the font size required for detection/discrimination at 75%.

I also have concerns with regard to the figures:

1) Figures 4 and 5 are not referenced in the text. All figures should be referenced in proximity to the results that they demonstrate.

2) The data figures are not as informative as they should be. This is mainly because the symbols and line styles are not explained. 2.1: What do the red/green lines/dots represent? If this distinction is important consider using colours that are accessible to a colourblind reader, or mark the category by varying symbol shape as well. 2.2: What does the dashed line represent in Figure 6?

Experimental design

This research is within the Aims and Scope of the journal, and the research question is meaningful and well defined. The role of the research in addressing holes in our current knowledge is clearly stated.

By my judgement, the psychophysical studies were conducted to a high technical and ethical standard. However, I believe there are gaps in the methods reporting that will preclude replication, and I have questions and suggestions with regard to the statistical models and presented data.

1) Methods reporting. Please include the following details:

line 116: ideally provide mean +/- SD for age and education level of participants in all experiments

line 120: what model of computer monitor? What refresh rate? Was monitor gamma linearised or native?

line 124: what was the viewing distance?

line 149: was the fade gaussian? linear ramping?

line 167: what was the monospace font? How large was the text displayed? I know you say later this was adaptive, but please indicate a range, and that the precise value was used to calibrate performance for each participant.

line 171: what was the range of the contrast used?

line 176: Please state explicitly that accuracy is the percentage of correct trials.

line 197: this section on how pupil size was converted from au to mm should be in the general methods, not the results. Also, it should include more detail. What was the function you determined?

line 237: Where were the peripheral tablets with respect to the participant? Please state in terms of distance and angle of visual field.

Validity of the findings

1) Comments on statistics

line 177: you say d' was similar to an accuracy based comparison. Please indicate where this can be verified (supplement? script in code/data package?).

line 260: There does not appear to be any hypothesis driving this on-the-fly measure of correlation between pupil size and threshold. I recommend removing it, and references to it through the paper.

line 261: Taking the mean of intermediate Quest estimates of threshold is not statistically sound. The procedure is intended to produce one estimate, informed by the responses to all previous trials (see Watson & Pelli, 1983 for an explanation of Quest).

2) Reporting within-subjects results and variability

line 178/179, line 200: please give mean and SD

Optional: consider making a much stronger test for within subject effects, and reporting within subject variability. I understand that you measure one accuracy measure per subject, per task, but you still have many trials with which to estimate variability (i.e. through bootstrapping.

3) Ergonomics
This paragraph is very difficult to follow and contributes very little ("notifications should be dark, except the text should be on a light screen, so maybe this is a dilemma for display design"), and deliberately ignores the other cues that make notifications detectable (they aren't at detection threshold, they have a sudden onset, they often move or bounce, and they are often accompanied by an auditory cue).

This needs to be either thought-out better and rewritten or removed from discussion and abstract.

Additional comments

Overall, I think this is a strong set of studies that need a moderate amount of work on the manuscript to be considered valuable, rigorous publications. Mainly, the manuscript needs to be expanded to include far more methodological detail, and the results need to be reported with more context, including the variability of measures of central tendency and improved integration between figures and text.

Other notes on clarity:

line 146: the description of the discrimination task sounds like a go/no-go change detection task. Please be clear whether you are using two-alternative forced choice (by context, it seems like 2afc is the method you are using).

Reviewer 2 ·

Basic reporting

The authors tackle an interesting question looking at the pupil size in relation to detection and discrimination performance. The language is professional and clear with detailed literature references. However there are a few improvements that must be addressed.

1) Some figures were not mention in the text at all! Figure 4 & 5 are included in the manuscript but not referred to in the text.

2) The "Discussion" are general summaries of results for each section. I would suggest to only use the Discussion heading for the "General Discussion".

Experimental design

The research question is well defined and falls within the scope and aims for this journal. The three experiments presented are defined and straightforward. However, there are a few issues with the analysis that needs to be clarified.

1) The statistical analysis (tests) for each experiment needs to be stated. "Conducted with the R package 1me4" isn't sufficient.

2) Why was results not normalised between subjects? If there is a good reason then it should be stated. Otherwise it seems that the difference in starting pupil size should mean the detection and discrimination would be affected due to the differences in light levels going into the eye. i.e. one with a starting pupil size of 2.5 mm compared to one at 4.5 mm.

3) How do you account for adaptation?

4) Have you considered testing with dilation drops (phenylephrine) and would you expect similar results?

Validity of the findings

The core question of this study is an interesting topic and should be visited. However the analysis needs to be clarified and in more detail in order to validate the results.

Reviewer 3 ·

Basic reporting

please see general comments

Experimental design

please see general comments

Validity of the findings

please see general comments

Additional comments

This is an important effort and certainly merits publication of all the reported results. There is considerable inaccuracy in recent pupil-related papers (notably, NOT from these authors), which often make unsupported claims on the impact of small pupil dilations/constrictions on visual performance - some taking for granted that larger pupil implies better vision, others taking for granted that smaller pupils imply better vision. Thus, the first important contribution of this paper is in its stating that increasing retinal illumination (e.g. by changing pupil diameter) do not have univocal consequences on visual performance: some visual tasks may improve, some may be hampered.

There are, however, aspects of the paper on which I have some criticism. I offer them in an attempt to further improve the positive impact of this paper on the community.


Main comments

The reported experiments attempt to evaluate the effect of pupil size by changing peripheral luminance, hence retinal illumination on contrast detection and acuity. This is a first very reasonable attack to the difficult question whether pupil size alone (irrespectively of luminance) can affect visual performance. However, it does have troubles that I would like to see discussed in more detail and addressed by additional analyses and revised figures:

1) plotting performance change as function of pupil size is misleading. The performance change is actually observed with a luminance drop of a factor of 10 and more. Achieving this drop by just changing pupil diameter would require a constriction of several millimeters: not the small changes that are reported on the x axis. For this reason, I suggest reporting both performance and pupil diameter as function of luminance (in addition to or in substitution of the current figures).

2) there are multiple reasons why visual acuity should improve with illumination. One is pupil size (as seen in Campbell and Gregory 1960), but this effect is minuscule compared with the change of contrast sensitivity and acuity that accompanies luminance increases and that is put down to properties of the retinal circuitry (here is my favourite picture for this line of work: https://webvision.med.utah.edu/wp-content/uploads/2018/05/KallSpat24.jpg). This factor should be discussed in the paper. Relatedly, I wonder whether there is any evidence that the improved visibility of black-on-white text (compared to white-on-black text) is partly dependent on pupil size. Is there any literature testing whether this advantage survives in conditions of Maxwellian view? My understanding of the work by N. Graham in the 70s is that such advantage for on- vs off- stimuli is still observed when pupil size is kept fixed (Maxwellian conditions), but I acknowledge that testing conditions used in those studies may not be directly relevant to the question presented here. Please comment.

3) given the above, and given the known effects of light scatter, it is expected that contrast detection and acuity should vary with retinal illumination. however, this is not the point of the paper, which is concerned with showing that pupil diameter has an effect on detection or acuity, in addition to or independent of the effect of luminance level.
One way to establish such independent contribution of pupil diameter is by looking at inter-individual variability: across subjects, illumination is kept constant, but pupil diameter may vary and this should result in across-subjects variations of acuity/detection performance. For experiment 3 (successfully measuring the acuity limit), such correlation is reported. However, the correlation is clearly influenced by an outlier (one out of the 20 subjects has both very low acuity and very large pupil diameter). Given the critical importance of this observation in the logic of the paper, please confirm that the correlation holds when this subject is excluded.
For the other experiments (where only the limit of detection performance could be adequately studied), there appears to be no such trend for correlating performance with pupil diameter.
what is the evidence that pupil diameter affects acuity/detection then?

Relatively minor comments

1) In the interest of precision, I would like the authors to clarify why they assume that larger pupils should improve detection by increasing light influx. I can see why increasing light influx improves absolute detection thresholds - the ability to catch single photons in scotopic conditions. I am not sure why increasing light influx should also improve contrast detection (which is tested in the reported experiments). The authors mention the concept of signal-to-noise ratio, but I don’t see how this should be affected (increasing light influx should equally affect both the signal and the noise, no?).

2) On the other hand, I think that increasing light influx should also increase light scatter, with a consequent detriment of contrast-detection performance. If so, perhaps the authors could stress that the two opposite effects will always combine, and whether performance increases or decreases depends on which of the two has the strongest effect. Considering this, I wonder whether the results from the contrast detection task could not actually be interpreted as evidence AGAINST pupil size affecting performance. If light scatter increases with pupil size, then detection should be hampered; however, he opposite is observed in both experiments 1 and 2.

3) It is increasingly common to read that pupil constriction improves visual acuity. Many authors use this to explain why pupil constrictions may accompany attention shifts, and some have gone so far as to suggest this as a possible confound that should be controlled for routinely, e.g. in EEG studies (Bombeke). In the face of this, another important contribution of this paper is to show how difficult/cumbersome it is to find an experimental condition where the acuity of central human vision can actually be modulated. Of three experiments, only the last one with very non-standard apparatus was successful in recording a measurable acuity change. I suggest that this point is emphasised further in the text.

---

## Round 0.2 · Minor Revisions

As you can see, both reviewers found the revised version to be an improvement, though both suggested very minor stylistic and grammar changes. Could you please review the submitted version and address the Reviewer's changes? I do not foresee a need to send the revised version for further review, as I do not wish to delay a final decision on this work any longer. Accordingly, if these amendments are incorporated, the revised version will require only editorial review (by me). Thank you again for choosing PeerJ as a forum for your work.

·

Basic reporting

Comment R1.5 - in the review guidelines, whether or not the article is written in "technically correct text" is not left as a matter of taste.

In the absence of a copyediting service at PeerJ, I recommend that the authors should revise which words they choose to capitalise to include only proper nouns, or those words at the beginning of sentences. Semantic importance is not grounds for capitalisation under any professional English standards.

Experimental design

no comment.

Validity of the findings

no comment.

Additional comments

The authors have done an excellent job integrating feedback into the manuscript, and I recommend it for acceptance.

Reviewer 3 ·

Basic reporting

no further comments

Experimental design

no further comments

Validity of the findings

The authors have addressed my concerns with additional analyses and further discussion.
However, the abstract and conclusions have been largely unaffected. My point was (and is) that luminance has very well known effects on visual sensitivity and acuity (and of course will change pupil size as well). The reported experiments manipulate luminance and find changes in visual sensitivity/acuity; relating these changes to pupil size is speculative, and should be presented as such. I agree with another reviewer that an effective way of manipulating pupil size independent of luminance, an thereby dissociate their effects, would have been a pharmacological intervention; however, I also agree with the editor that such investigation would go well beyond the scope of the present work. Thus, all I ask is that the authors rewrite parts of their text to clearly separate speculations (role of pupil size) from findings (effects of luminance). Small edits should suffice, for exampleP

abstract: "Small pupils lead to improved discrimination performance, consistent with the positive-polarity advantage" There is no evidence that small pupils *lead* to improved performance in these experiments. Rather, the performance improvement is observed at high luminance, and it is accompanied by small pupils. here, the only variations of pupil size independent on luminance are the inter-individual variations, and these do not correlate robustly with performance.

abstract & introduction: "The positive-polarity advantage is often linked to pupil size." the word "linked" suggests that a causal link between pupil size and positive-polarity advantage has been established by prior work, which is inappropriate. Perhaps the authors mean that the positive-polarity advantage is generally accompanied with small pupil size?

Similar instances are encountered throughout the text and in the conclusions, please rephrase

---

## Round 0.3 · accepted · Accept

Thank you for your careful attention to the Reviewers' comments and suggestions. I hope you will agree that their input resulted in valuable improvements to an already strong body of work. Congratulations, and thank you again for submitting your research to PeerJ.

Reviewer 3 ·

Basic reporting

nothing to add

Experimental design

nothing to add

Validity of the findings

nothing to add

Additional comments

nothing to add